# Evaluation of Polyphenol Intake in Pregnant Women from South-Eastern Spain and the Effect on Anthropometric Measures at Birth and Gestational Age

**DOI:** 10.3390/nu16183096

**Published:** 2024-09-13

**Authors:** Daniel Hinojosa-Nogueira, Desirée Romero-Molina, Beatriz González-Alzaga, María José Giménez-Asensio, Antonio F. Hernandez, Beatriz Navajas-Porras, Adriana Delgado-Osorio, Antonio Gomez-Martin, Sergio Pérez-Burillo, Silvia Pastoriza de la Cueva, Marina Lacasaña, José Ángel Rufián-Henares

**Affiliations:** 1Biomedical Research Center, Department of Nutrition and Bromatology, Institute of Nutrition and Food Technology, University of Granada, 18071 Granada, Spain; dhinojosa@ugr.es (D.H.-N.); beatriznavajas@ugr.es (B.N.-P.); adrianadelgado@ugr.es (A.D.-O.); spdelacueva@ugr.es (S.P.d.l.C.); 2Biosanitary Research Institute ibs.GRANADA, 18014 Granada, Spain; deromero@ugr.es (D.R.-M.); beatriz.gonzalez.easp@juntadeandalucia.es (B.G.-A.); mariajoseases@hotmail.com (M.J.G.-A.); ajerez@ugr.es (A.F.H.); antonio.gm.gr@gmail.com (A.G.-M.); 3Statistics and Operations Research Department, Faculty of Sciences, University of Granada, 18071 Granada, Spain; 4Andalusian School of Public Health (EASP), 18011 Granada, Spain; 5CIBER of Epidemiology and Public Health (CIBERESP), Instituto de Salud Carlos III, 28029 Madrid, Spain; 6Department of Legal Medicine and Toxicology, University of Granada, 18016 Granada, Spain; 7Department of Pharmacology and Pediatrics, University of Málaga, 29010 Málaga, Spain; spburillo@ugr.es

**Keywords:** foetal anthropometry, polyphenols, pregnant women, 24-h dietary recalls, food frequency questionnaire

## Abstract

During pregnancy, controlling nutrition is crucial for the health of both mother and foetus. While polyphenols have positive health effects, some studies show harmful outcomes during pregnancy. This study evaluated polyphenol intake in a cohort of mother–child pairs and examined its effects on foetal anthropometric parameters. Polyphenol intake was assessed using food frequency questionnaires (FFQs) and 24-h dietary recalls, and analysed with the Phenol-Explorer database. Gestational age and birth measurements were retrieved from medical records. Statistical analyses validated dietary records and assessed polyphenol impact using multivariate generalised linear models. The study found that mean gestational age was 39.6 weeks, with a mean birth weight of 3.33 kg. Mean total polyphenol intake by FFQ was 2231 mg/day, slightly higher than 24-h recall data. Flavonoids and phenolic acids constituted 52% and 37% of intake, respectively, with fruits and legumes as primary sources. This study highlights the use of FFQs to estimate polyphenol intake. Furthermore, the study found associations between polyphenol consumption and anthropometric parameters at birth, with the effects varying depending on the type of polyphenol. However, a more precise evaluation of individual polyphenol intake is necessary to determine whether the effects they produce during pregnancy may be harmful or beneficial for foetal growth.

## 1. Introduction

Balanced maternal nutrition during pregnancy is a pivotal area of study, as it exerts a significant influence on foetal growth and the overall well-being of both mother and offspring [1,2]. Numerous epidemiological studies suggest that diets rich in plant-based foods have long-term beneficial effects on both the foetus and maternal health [3,4]. Among the compounds contributing to these effects are polyphenols, with over 8000 molecules so far [5]. These compounds are secondary metabolites derived from plants and play a crucial role in their survival and adaptation. Phenolic compounds can be classified into five broad categories: flavonoids, phenolic acids, lignans, stilbenes and other polyphenols [6,7,8]. The primary dietary sources of polyphenols include vegetables, fruits, cereals, nuts, legumes, chocolate and similar foods [9,10]. Due to their beneficial effects on health, dietary polyphenols have received considerable attention. They are primarily recognised for their antioxidant activity, which is associated with other beneficial properties such as modulation of inflammatory responses, anti-obesogenic activity and reduction of the risk of cardiovascular disease [3,10,11,12,13,14]. Additionally, polyphenols have been linked to neuroprotective functions [15,16,17] and the inhibition of tumour growth in various types of cancers, including those affecting the colon, prostate and breast [11,12,13]. Given their health benefits, there is considerable public interest in increasing polyphenol intake through diet, nutraceuticals, fortified foods, beverages and dietary supplements [18].

Although the beneficial effects of polyphenols are well documented, concerns have been raised about the potential health risk of these compounds [9,14,19,20,21,22,23]. At present, the toxicity of polyphenols remains uncertain, but the availability of polyphenol-rich foods and supplements is growing. Amid increasing concern, a database named ToxDP2 has been created [21], providing comprehensive data on 415 dietary polyphenols that may have toxicological effects. In addition, studies suggest that consumption of large amounts of polyphenols may have pro-oxidant effects. They have also been associated with hepatotoxicity or an increased risk of certain types of cancer [10,14,19,21,24,25]. One of the major risks associated with polyphenol consumption is the constriction of the foetal ductus arteriosus during the third trimester of pregnancy [9,19,20,21,23,26,27,28,29,30,31,32,33,34,35], which may lead to potentially serious consequences, including perinatal pulmonary hypertension, heart failure and foetal death [20,28]. Therefore, understanding the relationship between polyphenol consumption and its effects on maternal and foetal health is essential.

Commonly, methods used to evaluate dietary polyphenol consumption during pregnancy involve the use of dietary records such as food frequency questionnaires (FFQs) [30,36,37]. In addition, accurate quantification requires a database containing polyphenol concentrations in various foods. Phenol-Explorer is one of the most widely used databases for this purpose [8]. Considering the above, this study aimed to evaluate polyphenol consumption in a cohort of pregnant women from South-eastern Spain and its potential effects on foetal health, focusing on anthropometric measurements at birth.

## 2. Materials and Methods

### 2.1. Patient Selection and Study Design

This prospective, population-based, pregnant-women birth cohort study, known as the GENEIDA Project, “Genetics, early life environmental exposures and infant development in Andalusia” (https://www.easp.es/web/geneida/, accessed on 12 August 2024), started in 2014 in a well-defined geographic area of South-eastern Spain (Almería). Inclusion criteria for enrolment included pregnant women aged 17 years or older who intended to give birth at the referral hospital. Additionally, participants were required to have singleton pregnancies (not resulting from assisted reproductive technology) and no pre-existing chronic diseases. Furthermore, they could not currently be receiving medical treatment, and could not have any language barriers.

Women were enrolled in the study at their first antenatal visit at the hospital (around 12–13 weeks of gestation) and were followed throughout pregnancy, delivery and after birth at different children’s development ages (1, 2, 4 and 7–8 years). A total of 800 women were recruited, and the final analysis was based on 680 women (85%). Pregnant women completed the same FFQ on two occasions, initially during the first trimester and subsequently during the third trimester. Simultaneously, 24-h dietary recalls were conducted in a subsample of 40 women for this specific study.

This study was approved by the Biomedical Research Ethics Committee of Andalusia, and all participants provided written and informed consent. The study objectives were clearly explained and participants had the right to withdraw from the study and request withdrawal of their data at any time. Throughout the study, we adhered to the ethical principles of the World Medical Association’s Declaration of Helsinki, as well as the ethical and legal standards of Spanish legislation.

### 2.2. Dietary Assessment

A variety of dietary records were administered to study participants. A semi-quantitative FFQ previously validated for assessing total nutrient intake was used for dietary recording [38]. The FFQ covered 141 items, which were further classified into 28 food subgroups. These items included specific traditional dishes, spices and foods most frequently consumed in the study area. Respondents had nine options to choose from, ranging from “never” or “almost never” to “more than six times a day”. The 24-h dietary recall method was selected as the reference system to validate the FFQ for the polyphenol intake [30]. This validation was performed in a subsample of the study population. The data from this subsample were collected from those participants who volunteered to take part. All foods and beverages consumed in the previous 24-h dietary recalls were compiled over three non-consecutive days, including two weekdays and one weekend day.

Both dietary records were completed twice during pregnancy: the first covered the time before the first trimester of pregnancy and the second during the third trimester. To enhance response rates and data accuracy, trained interviewers assisted participants in completing the questionnaires, thereby minimising bias.

### 2.3. Quantitative Estimation of Total Polyphenol Intake

The Phenol-Explorer 3.6 database was used to estimate polyphenol intake. This resource provided data on the total polyphenol content of foods using the Folin–Ciocalteu method. Additionally, we determined the concentrations of individual polyphenol families and subfamilies [8]. The mean total polyphenol intake (MTPI) of each participant was estimated from the mean of the Folin–Ciocalteu total polyphenol content and the sum of the concentrations of each polyphenol subfamily. The subfamilies of phenolic compounds were categorised into several main groups based on Phenol-Explorer criteria: (a) flavonoids (including anthocyanins, chalcones, dihydrochalcones, dihydroflavonols, flavanols, flavanones, flavones, flavonols and isoflavonoids); (b) phenolic acids (such as hydroxybenzoic acids, hydroxycinnamic acids, hydroxyphenylacetic acids and hydroxyphenyl propanoic acids); (c) lignans (lignans); (d) stilbenes (stilbenes); and (e) other polyphenols (alkylmethoxyphenols, alkylphenols, furanocoumarins, hydroxybenzaldehydes, hydroxybenzoketones, hydroxycinnamaldehydes, hydroxycoumarins, methoxyphenols, naphtoquinones, tyrosols and other polyphenols). Some subfamilies were transformed into dichotomous variables due to the large number of participants who did not consume foods containing these polyphenols. The foods were classified into eleven groups: cereals and derived products, vegetables, fruits, legumes, nuts, oils, fruit derivatives (e.g., fruit juices), chocolate and coffee, spices and infusions, alcoholic beverages and processed foods (such as pizzas, lasagnas, etc.). Processed foods were separated according to their main ingredients following typical commercial recipes and were adjusted using cooking yield factors to estimate the total phenols of that food in the diet [6,7,8]. Any food lacking polyphenols was excluded from the study. For the estimation of polyphenols in 24-h dietary recalls, a previously developed and validated tool was used [39].

### 2.4. General Questionnaire, Medical Records and Anthropometric Measurements

Information from participants was collected using a structured questionnaire administered by trained staff during their hospital appointment scheduled for the first and third trimesters of pregnancy. The questionnaire covered sociodemographic characteristics, working life and occupational exposure, exposure at home and living environment, obstetric history and prior illnesses. Pregnant women self-reported relevant data such as pre-pregnancy weight and height during the first trimester of pregnancy. Weight at 32 weeks’ gestation was extracted from medical records. Body mass index (BMI) was calculated from height and weight using the function (kg/m^2^). The gestational age of the pregnancy, defined as the time elapsed between the first day of the last menstrual cycle and the time of delivery, along with anthropometric measurements, including birth weight, height and head circumference, was obtained from hospital records in accordance with relevant guidelines and standardised protocols [40,41,42]. The ponderal index was calculated using the following formula: weight (g) × 100/(length, cm)^3^. Specific z-scores for weight, length and head circumference at birth were calculated. These adjustments are described in more detail in previous studies [43].

### 2.5. Data Analysis

Statistical data were analysed using the SPSS statistical package version 26.0, R software version 4.3.2 and Python version 3.7. Bivariate and multivariate generalised linear models (GLMs) were used to evaluate the impact of polyphenol intake on height z-score, weight z-score, head circumference (HC) z-score and ponderal index. The mother’s polyphenol intake during the first and third trimesters of pregnancy was used as the independent variable, and all models were adjusted for energy intake to ensure accurate estimation. The following confounders and covariates were identified based on previous studies and considered for adjustment in the multivariate models: energy intake, family’s monthly income, gestational weight gain, gestational age at the first prenatal visit, infant sex, marital status, maternal education, maternal stress, mother’s age, physical exercise, pre-pregnancy BMI, history of repeated abortions, rural/urban residence, supplement intake, type of delivery, alcohol consumption, ethnicity, parity, season and smoking. Variables were mapped and their relationships analysed using a directed acyclic diagram (DAG) generated by DAGitty version 3.0 software (Figure 1).

With this information, and using a stepwise method of variable selection, the following confounding factors were finally selected for inclusion in multivariate models: height z-score (smoking, alcohol intake and repeated abortions); weight z-score (smoking, repeated abortions, maternal education and parity); head circumference z-score (smoking, mother’s age, maternal physical exercise, supplement intake, education and parity), and ponderal index (smoking, mother’s age, maternal education and parity).

Means and standard deviations (SD) for all polyphenol intakes were obtained for the 24-h dietary recalls and the two FFQs. Previously, the validity and reproducibility of the polyphenol intake FFQ were assessed using statistical approaches, in line with the methodology previously used for nutrients [38]. Briefly, the methodologies commonly used for validation of nutritional parameters include the correlation coefficient, quintile ranking and limits of agreement (LOA). In particular, Spearman’s correlation coefficient was determined for MTPI according to the distribution in the different food groups. In quintile ranking, polyphenol intake was divided into quintiles and the percentage of data correctly classified in the same or adjacent quintiles was calculated. The LOA technique, or Bland–Altman method, is a graphical technique where the limits of agreement are established as ±1.96 SD of the mean difference between the polyphenol intakes obtained from two questionnaires. In this technique, the percentage of data falling within these graphical limits was counted. The significance level was set at *p* < 0.05.

## 3. Results

### 3.1. Characteristics of the Study Population

The characteristics of 680 pregnant women and their newborns from the GENEIDA cohort are shown in Table 1.

The mean maternal age was 31 years with a standard deviation of 4.8 years old. The mean height of the participants was 1.64 m and pre-pregnancy weight 64.7 kg. The pre-pregnancy BMI was 24.2 ± 4.5 kg/m^2^. The mean gestational age was 39 weeks with a standard deviation of 1.3 weeks and 48.7% of the newborns were girls. Regarding the anthropometric characteristics of the newborns, the mean birth weight was 3.33 kg, the mean birth length 50.7 cm, the head circumference 33.8 cm and the ponderal index 2.55.

### 3.2. Validity and Reproducibility of FFQ

The validation study used dietary information provided by 40 out of the 680 pregnant women participating in the study. These women completed all the FFQ and 24-h dietary recalls. The correlation coefficients for MTPI according to distribution in the different food groups ranged from 0.42 (for fruit polyphenols) to 0.02 (for legume polyphenols). In the case of mean total polyphenol intake, a statistically significant (but low) correlation coefficient of 0.3 was found. According to quintile classification, polyphenol intakes for each food group in the same (or adjacent) quintile ranged from 76.7% to 53.5% for the groups “chocolate and coffee” and “oils”, respectively (Table 2). The limits of the agreement varied from 95.4% to 90.7%.

For reproducibility, FFQs during the first and third trimesters of pregnancy were compared. The correlations between the two FFQs regarding the contribution of MTPI and the polyphenol families are shown in Table 3. Comparisons were also made between the different food groups. Correlation coefficients ranged from 0.52 to 0.32 for the vegetable and cereal groups, respectively. In relation to the polyphenol intake, the ranges varied from 0.141 for stilbenes to 0.406 for phenolic acids. For MTPI, the polyphenol families and all food groups had significant correlations with *p* < 0.01. The percentage of food groups classified in the same quintile by the two FFQs ranged from 74.5% for the group of vegetables to 65% for the fruit group. In the case of polyphenol families, the mean values classified in the same quintile were 66.2%. The limits of agreement for all values were distributed over 94.7% (Table 3).

### 3.3. Total Polyphenol Intake

The MTPI for the 680 women during pregnancy was 2231 ± 757 mg/day calculated on the basis of information provided by the FFQs, and 1875 ± 835 mg/day from the 24-h dietary recalls (40 pregnant women). The total polyphenol consumption estimated from the FFQs was slightly higher when compared with the average of the 24-h dietary recalls. Total polyphenol content of food calculated by the Folin–Ciocalteu method was 3367 ± 1167 mg/day, while 1069 ± 421 mg/day was obtained with the sum of the concentrations of the individual polyphenol subfamilies. The flavonoid group was the most abundant polyphenol family, accounting for about 52% of the total intake. Phenolic acids accounted for 37% of the total, making them the second-most abundant group. Lignans and other polyphenols each represented about 5%, while stilbenes were the minority group. If we focus on subfamilies, flavanols (33%), hydroxycinnamic acids (21%), flavanones and anthocyanins are the most representative, accounting for 73% of the total intake. The results by subfamily of phenolic compounds are shown in the Appendix A.

Of the 11 food groups, legumes (28.5%), fruits (25.3%), vegetables (15.2%) and chocolate and coffee (14.1%) were the major dietary sources of total polyphenols in the diet of the GENEIDA cohort. Within each food group, the main contributors were identified and were consistent with the 24-h dietary recalls. Lentils, cocoa powder, apples, chocolate, oranges, tomatoes, gazpacho and capsicum were the foods with the highest contribution to dietary polyphenols in relation to total polyphenol intake and regular food consumption.

### 3.4. Relationship between Polyphenol Intake and Anthropometric Measures at Birth

Bivariate and multivariate GLMs were used to evaluate the potential effects of polyphenol intake (total, by families or subfamilies) on different birth anthropometric measures (height z-score, weight z-score, head circumference z-score and ponderal index) at different trimesters of pregnancy and the average of the two calculated (Appendix A).

Figure 2 presents the resulting regression coefficients of polyphenols (total and families) in the multivariate models, including their confidence intervals. Beta coefficients indicate that polyphenols significantly increased birth anthropometric measures for some of the polyphenol families at some gestational periods. The intake of phenolic acids during the third trimester, which accounted for over 30% of the total intake, was significantly associated with increased head circumference at birth. Also, phenolic acids showed a near-significant positive relationship with weight during the first trimester and on average. Lastly, stilbenes exhibited a significant positive association with both height and weight at birth in the third trimester.

An in-depth analysis, taking into account the subfamilies of polyphenols, revealed significant associations (Figure 3). Statistically significant inverse associations were found, particularly during the first trimester of pregnancy, between the ponderal index and intake of hydroxyphenyl acetic acids, as well as between hydroxyphenyl propanoic acids and tyrosols. However, significant direct associations were observed between different subfamilies of polyphenols and anthropometry at birth measures, with the exception of dihydroflavonols. Only some of the positive direct associations remained when considering average pregnancy intake and anthropometric measures at birth.

## 4. Discussion

### 4.1. Validation and Reproducibility

In the current study, carried out in a population of healthy pregnant women in the south of Spain, the intake of total polyphenols was evaluated using various dietary assessment tools. Although the FFQ used is a valid and reproducible tool for assessing nutrients [38], it is essential to demonstrate its validity and reproducibility specifically for polyphenol intake. Although FFQs are widely used to estimate total polyphenol intake, only a few have been adequately validated [36,44]. Furthermore, finding validated tools specifically for estimating polyphenol intake in pregnant women poses additional challenges [30]. Our study included correlation coefficients, percentages of LOA, and the percentage of agreement by quintiles (as shown in Table 2 and Table 3). The results obtained were comparable to the values observed for the validation of other nutrients [38]. Similarly to the validation of different macro and micronutrients, the correlation coefficients for validation analysis of polyphenol intake were low, while the rest of the statistical tests yielded optimal results. These correlation coefficients were comparable to those reported by other studies that validated different FFQ in pregnant women [45]. Interestingly, the reproducibility analysis revealed higher correlation coefficients than those obtained from the validation (Table 2 and Table 3). This is consistent with the findings of other FFQ validation and reproducibility studies. Overall, the results indicate an acceptable level of validity and high reproducibility for all food groups and polyphenol intakes during pregnancy. This was evidenced by a percentage of agreement by quintiles higher than 60% and LOA exceeding 90% in both cases that was consistent with other studies [17,29]. Therefore, the use of this FFQ represents a valuable tool for the estimation of polyphenol intake during pregnancy.

### 4.2. Polyphenol Intake

The total consumption of polyphenols, around 2 g/day, is consistent with findings from other studies [12,44,46,47]. In the present study, statistically significant differences were observed between total polyphenol intakes obtained by the FFQ and 24-h dietary recalls for vegetables, spices and infusions, legumes, nuts, processed foods and oils (Table 2). The FFQ revealed higher contributions from legumes, while the 24-h diet recall indicated greater contributions from spices and infusions. These discrepancies underscore the importance of detailed recording to accurately assess the true contribution of polyphenols from certain foods such as spices and infusions [48], which gives consistency to the results found. Given that FFQs often inadequately cover spice and infusion food groups, the current trend involves incorporating them into new FFQs that are undergoing validation [49].

Looking closely at the data from our study, foods with the highest dietary polyphenol content were also identified as sources of polyphenols in other studies [50,51]. Among populations with the highest polyphenol intake, legumes were the primary food group contributing to daily polyphenol intake. Recent studies have emphasised the significant role of legumes, accounting for up to 32% of total dietary polyphenol intake [52]. Cocoa and chocolate were additional sources of polyphenol intake in women of the GENEIDA cohort, aligning with findings from previous studies [53].

When comparing the average intake of food groups during the first and third trimesters of pregnancy, a general decrease in consumption across various foods was observed. Specifically, there was a lower intake of processed foods, chocolate, coffee and alcoholic beverages. Such a decrease may be attributed to heightened awareness of maintaining a healthy diet during pregnancy [54].

Flavonoids, phenolic acids and lignans were the families that contributed most to daily polyphenol intake, as also reported by previous studies [11,55]. Similar results were found for the subfamilies of phenolic compounds [11,56].

The limited data available on polyphenol consumption in pregnant women make comparisons difficult. In a comprehensive cohort study involving 120 pregnant Brazilian women, the average polyphenol intake was 1048 ± 362 mg/day [30]. This amount is similar to the sum of the concentrations of individual polyphenol subfamilies found in our study, although both exceeded the levels reported for pregnant women in China [57]. Additionally, a study validating an FFQ in the general population (aged 20–60 years) revealed polyphenol intake similar to that observed in the GENEIDA cohort (2111 mg/day) [58].

Polyphenol intakes among European women vary between 653 and 1552 mg per day [25]. For example, the cohort of women in the HAPIEE (Health, Alcohol and Psychosocial factors In Eastern Europe) had an average polyphenol intake of 1726 ± 662 mg/day [59]. In the GENEIDA cohort, fruits, vegetables and cocoa products were major contributors to daily polyphenol intake. There are notable differences in the consumption of polyphenols between different parts of the world. These variations may stem from differences in population characteristics, dietary behaviours, or the tools and databases employed to assess food intake [59].

Most academic sources evaluate polyphenol intake based on individual polyphenols or families, but few combine the results to estimate total intake, which may provide a less realistic estimate [56]. Furthermore, it is important to consider the databases used because although some researchers prefer to use their individual databases, Phenol-Explorer tends to be one of the most commonly used options [60,61]. Despite Phenol-Explorer being one of the most widely used approaches, polyphenol content databases have limitations and may underestimate intake. The MTPI was devised for this reason, aiming to supplement overall values rather than merely summing up individual polyphenols. A clear example of daily polyphenol intake in the Spanish population illustrates this variation. Records show values ranging from 671 mg/day [25] to as high as 2590–3016 mg/day [37]. The difference lies in whether the approximations include extractable or non-extractable polyphenols, which depends on the dietary records and databases used.

### 4.3. Relationship between Prenatal Polyphenol Intake and Birth Anthropometry and Foetal Development

Some studies suggest that a high intake of polyphenols may impair foetal development and contribute to different problems such as low birth weight [19,20,21,23,26,27,28,29,31,32,33,34,35]. These studies are based on the properties of polyphenols similar to anti-inflammatory drugs, which may be harmful during foetal development by interacting with the foetal ductus arteriosus [32].

The ductus arteriosus is an essential structure in foetal circulation, connecting the pulmonary artery with the aortic arch during foetal life. It begins to close within the first hours after birth, becoming part of the adult circulation pattern by 72 h [28]. Foetal ductus arteriosus constriction is a clinical disorder caused by inhibition of the prostaglandin synthesis pathway, and has long been associated with maternal intake of nonsteroidal anti-inflammatory drugs in late pregnancy [32]. Over the years, researchers have studied the potential association between polyphenol intake and constriction of the foetal ductus arteriosus, which, although rare, is a condition often considered idiopathic [32]. The effects of polyphenols on ductal dynamics have been well documented in animal studies [35], and include reduced litter size, foetal head circumference and foetal abdominal circumference in mice [27]. Evidence from several studies supports a cause–effect relationship between maternal consumption of polyphenol-rich substances (such as herbal teas, orange and grape juice, chocolate and cocoa) and constriction of the foetal ductus arteriosus [23,28,31,33,34,35]. Recommendations to prevent foetal ductal constriction during the third trimester of pregnancy have been subject to debate, and include possible dietary modifications to reduce tea, chocolate or cocoa consumption [28]. As part of these guidelines, a low-polyphenol diet has been proposed for women with foetal ductal constriction. Notably, the majority of foetuses receiving this dietary intervention showed reversal after a three-week period of low-polyphenol intake, suggesting the effectiveness of this intervention [23,31,33].

Similar ductal problems have been identified [19]. For example, cases of premature closure of the ductus arteriosus have been associated with maternal consumption of functional foods with high anthocyanin content [29], or excessive tea consumption [26]. After identifying possible causes, pregnant women were advised to reduce the consumption of these foods and, at the end of the dietary intervention, a progressive improvement of ductal constriction was observed [26]. Furthermore, other harmful consequences have been documented, including an increased likelihood of neural tube defects in a Chinese population that regularly consumed tea [62].

However, there is certain discordance, as other studies found no significant harmful effects after conducting similar research. Despite high levels of hydroxytyrosol supplementation in pigs, no effect on ductus arteriosus constriction was observed during pregnancy [15,63]. On the other hand, certain studies have shown beneficial effects for pregnant women, including improvements in blood pressure and reductions in gestational diabetes [3,57].

In this study, the intake of polyphenols in the GENEIDA cohort was estimated to evaluate whether the intake of these chemical species is associated with anthropometric measures at birth. Results indicate that foetal growth can be influenced by certain types of polyphenols in varying ways, and the observed trends underscore the importance of considering not only the total intake of polyphenols but also the specific types consumed. This approach is consistent with the fact that polyphenols can function as both antioxidants and pro-oxidants, depending on their structure and concentration [24].

The findings of our study indicate that the effect of polyphenols on pregnancy varies not only with the specific type of polyphenol, but also with the trimester of pregnancy during which exposure occurs (Appendix A). This suggests that the physiological changes occurring in the mother throughout pregnancy may influence how different polyphenol structures affect the course of pregnancy and foetal development.

Despite the results obtained, further research is needed to explore the effect of polyphenols on foetal development, as suggested by other studies [22], especially by studying specific subfamilies [21,24]. In light of the results obtained, it is possible to see how families of polyphenols such as tyrosol or other polyphenols can have certain negative associations with anthropometric parameters at birth. Similarly, this phenomenon can be observed within certain subfamilies, such as certain phenolic acids. On the other hand, our study indicates that lignans can significantly affect anthropometric parameters in a positive manner. Our results are also consistent with previous studies that have found beneficial effects of lignans intake in pregnant women [64]. However, other studies have examined the oestrogenic effects of these compounds and their potential impact on pregnancy, highlighting the need for further studies to ensure their safety [65]. For these reasons, precaution should be taken during the latter stages of pregnancy, when it is advisable to limit the consumption of foods rich in polyphenols. For instance, in Brazil, guidelines for foetal cardiovascular health recommend limiting the intake of foods with a high polyphenol content during the last three months of pregnancy [28].

The significance of this issue lies in the fact that numerous dietary supplements and nutraceuticals currently consumed contain high levels of polyphenols. For instance, individuals who take supplements may consume up to 100 times more polyphenols daily [18]. While this heightened intake could be beneficial for health, some harmful effects may occur during specific life stages, such as pregnancy. Hence, careful control of polyphenol intake during gestation is necessary.

### 4.4. Strengths and Limitations of the Present Study

While the FFQ may overestimate dietary polyphenol intake, it is important to note that our results were estimated from a large population-based cohort of pregnant women. Additionally, the validity and reproducibility of the tools used for assessing polyphenol intake are important aspects to consider. Limitations of using databases should also be considered; although Phenol-Explorer is widely used at present, perhaps the inclusion of additional information could complete the results. Despite the remarkable results obtained, the associations found, although significant, remain somewhat inconclusive. This study is one of the few to date that highlight the importance of distinguishing between the various types of polyphenols when making recommendations of dietary restrictions during pregnancy, as some of them may be beneficial for foetal development while others may have adverse effects.

## 5. Conclusions

This study highlights the importance of comprehensively evaluating polyphenol intake during pregnancy, due to its impact on anthropometric measurements at birth. The results indicate that a validated FFQ can be an effective and reliable tool for estimating polyphenol consumption in pregnant women. In the GENEIDA cohort, legumes, fruits, vegetables, chocolate and coffee were the primary dietary sources of total polyphenols. The findings suggest that the intake of different polyphenol families and subfamilies may have diverse effects on foetal development. Some polyphenols may have beneficial effects, while others could be harmful. This underscores the importance of considering not only the overall quantity of polyphenols consumed, but also the specific classes of polyphenols ingested. Further research is required to elucidate the impact of individual polyphenol intake on maternal and foetal health during pregnancy.

These findings are crucial for the formulation of more precise dietary guidelines and recommendations during pregnancy to ensure optimal neonatal development.

## Figures and Tables

**Figure 1 nutrients-16-03096-f001:**
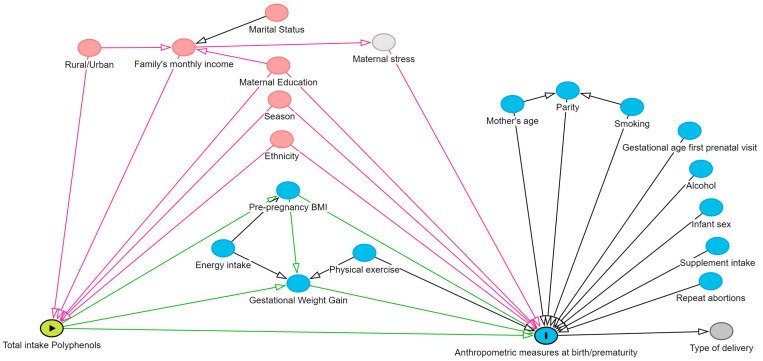
Direct acyclic graph (DAG) that displays the potential relationship between the variables and covariates considered for this study. "Triangle" indicates the exposure variable and "I" is the outcome. Red circles: confounding variables. Blue circles: covariates (causality associated with health outcomes). Light grey circle: variable not available in our study. Dark grey circle: descendant variable. Green arrows: causal path. Pink arrows: biasing path.

**Figure 2 nutrients-16-03096-f002:**
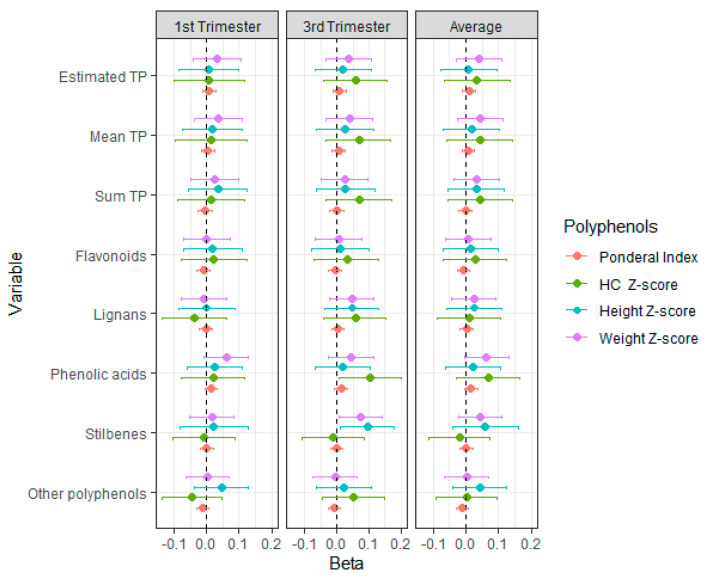
Associations between sum of total polyphenol intake and by family groups and anthropometric measures at birth. GLM adjusted for height z-score (smoking, alcohol and repeat abortions); weight z-score (smoking, repeat abortions, maternal education and parity); head circumference (HC) z-score (smoking, mother’s age, maternal, physical exercise, supplement intake, education and parity); and ponderal index (smoking, mother’s age, maternal education and parity). Regression coefficients and 95% confidence intervals are presented. TP: total polyphenols.

**Figure 3 nutrients-16-03096-f003:**
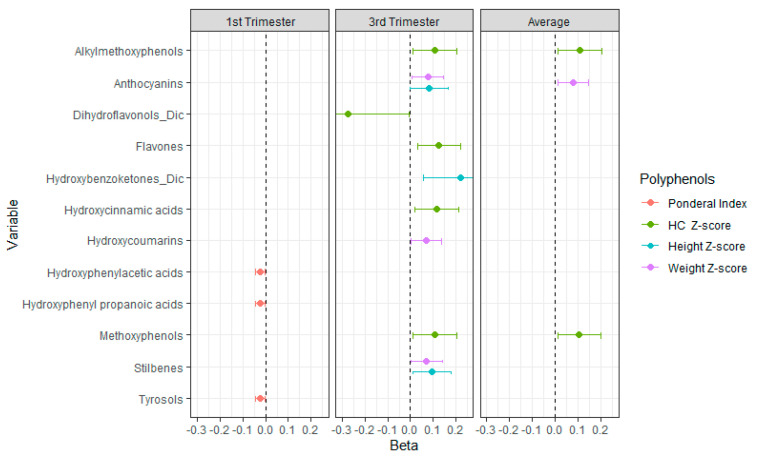
Associations between intake of polyphenols by subfamilies during pregnancy and anthropometric measures at birth (only statistically significant associations are shown). GLM adjusted for height z-score (smoking, alcohol and repeat abortions); weight z-score (smoking, repeat abortions, maternal education and parity); head circumference (HC) z-score (smoking, mother’s age, maternal, physical exercise, supplement intake, education and parity); and ponderal index (smoking, mother’s age, maternal education and parity). Regression coefficients and 95% confidence intervals are presented. Figure presents the results of the models that had statistically significant effects.

**Table 1 nutrients-16-03096-t001:** Demographic characteristic of the study population (*n* = 680).

Characteristics	No. (%), Mean ± SD
**Maternal characteristics**	
Maternal age	31.05 ± 4.86
Education:	
Primary education	335 (49.3%)
Secondary education	162 (23.8%)
Higher education	183 (26.9%)
Parity:	
0 (primiparous)	262 (38.5%)
≥1 (multiparous)	418 (61.5%)
Repeat abortions:	
Yes	54 (7.9%)
No	626 (92.1%)
Smoking:	
Never	569 (83.6%)
Only 1st trimester	29 (4.3%)
During all pregnancy	82 (12.1%)
Alcohol consumption:	
Never	133 (19.5%)
Only 1st trimester	282 (41.5%)
During all pregnancy	265 (39.0%)
Vitamin supplement intake:	
Never	538 (79.1%)
Sometime during pregnancy	129 (19.0%)
During all pregnancy	12 (1.8%)
Physical exercise:	
Never	14 (2.1%)
Sometime during pregnancy	138 (20.3%)
During all pregnancy	528 (77.6%)
Energy intake (Kcal):	
1st trimester	2404 ± 739
3rd trimester	2054 ± 670
Pre-pregnancy BMI (kg/m^2^)	24.23 ± 4.68
Gestational weight gain (kg)	11.23 ± 5.42
**Infant characteristics**	
Infant sex:	
Male	349 (51.3%)
Female	331 (48.7%)
Length of gestation (weeks)	39.87 ± 1.25
Ponderal index	2.55 ± 0.25
Birth weight (g)	3329.30 ± 427.05
Birth length (cm)	50.70 ± 2.09
Head circumference (cm)	33.75 ± 1.52

**Table 2 nutrients-16-03096-t002:** Validation of daily intake of total polyphenols based on food frequency questionnaire (FFQ) and 24-h dietary recalls (*n* = 40).

	FFQ 1	FFQ 2	Correlation Coefficient	Agreement by Quintiles (%) ^a^	Agreement by LOA (%) ^b^
	Mean ± SD	Mean ± SD			
Mean total polyphenol intake (mg)	2158 ± 1023	1875 ± 835	0.303 *	67.4	95.4
Vegetables (mg) ^†^	476 ± 321	312 ± 296	0.369 *	62.8	93.1
Spices and infusions (mg) ^†^	7.33 ± 5.02	139 ± 209	0.326 *	67.4	95.4
Cereals and derived products (mg)	230 ± 131	234 ± 126	0.314 *	69.8	90.7
Legumes (mg) ^†^	676 ± 408	219 ± 395	0.015	60.5	93.1
Fruits (mg)	358 ± 185	407 ± 279	0.418 **	69.8	90.7
Fruit derivatives (mg)	68.5 ± 30.9	49.3 ± 81.5	0.224	62.8	90.7
Oils (mg) ^†^	29.5 ± 15.8	12.3 ± 9.57	−0.086	53.5	95.4
Nuts (mg) ^†^	45.9 ± 10.4	33.4 ± 68.8	0.261	67.4	95.4
Processed foods (mg) ^†^	66.8 ± 49.3	65.1 ± 81.1	−0.056	67.4	90.7
Chocolate and coffee (mg)	418 ± 345	564 ± 527	0.347 *	76.7	93.1
Alcoholic beverages (mg)	5.07 ± 2.73	4.87 ± 9.01	0.260	65.1	90.7

^a^ Correctly classified if classified into same or adjacent (±1) quintiles. ^b^ Overall proportion of agreement limits between both questionnaires, corresponding to Bland–Altman plots. * Correlation significant at *p* < 0.05 level; ** Correlation significant at *p* < 0.01 level. ^†^ Significant differences (*p* < 0.05), paired-sample sign test, observed between total polyphenol intakes obtained by FFQ and 24-h dietary recalls.

**Table 3 nutrients-16-03096-t003:** Reproducibility analysis of polyphenol intakes (mg) based on two food frequency questionnaires (FFQ), during first and third trimesters of pregnancy (*n* = 680).

	FFQ 1	FFQ 2	Correlation Coefficient	Agreement by Quintiles (%) ^a^	Agreement by LOA (%) ^b^
	Mean ± SD	Mean ± SD			
Mean total polyphenol intakes	2388 ± 905	2075 ± 932	0.355 **	65.2	94.1
Flavonoids	624 ± 364	518 ± 353	0.336 **	69.8	93.5
Phenolic acids	461 ± 225	350 ± 200	0.406 **	65.6	95.7
Lignans	68.3 ± 45.4	57.8 ± 45.7	0.376 **	68.9	94.4
Stilbenes	0.47 ± 0.53	0.2 ± 0.24	0.141 **	60.1	96.6
Other polyphenols	60.5 ± 46.9	50.6 ± 40.9	0.321 **	67.5	94.1

^a^ Correctly classified if placed into the same or adjacent (±1) quintiles. ^b^ Overall proportion of agreement limits between both questionnaires, corresponding to Bland–Altman plots. ** Correlation significant at *p* < 0.01 level.

## Data Availability

The data presented in this study are available on request from the corresponding author. The data are not publicly available due to confidentiality concerns.

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
