# Peer review of "Evaluation of Polyphenol Intake in Pregnant Women from South-Eastern Spain and the Effect on Anthropometric Measures at Birth and Gestational Age"

_nutrients, 2024, doi:10.3390/nu16183096_

Round 1
Reviewer 1 Report
Comments and Suggestions for Authors
Dear Authors,
Thank you for the opportunity to review this paper. The study deeply analyses the relationship between polyphenols intake during pregnancy and infant anthropometric measurement. The paper is well written and the statistical methods are robust.
I congratulate the authors for this complex study, including the validation of the FFQ.
Please find some comments below:
Line 94 The women completed the two FFQs in the first and third trimesters). It is unclear whether the two FFQs are different? Why?
Line 137 The foods are classified in groups> cereals and carbohydrates. It is an inappropriate group because cereals are foods and carbohydrates are macronutrients. Please replace somehow.
Line 152-155 It is unclear. The gestational age and anthropometric measurement for newborns…. Please clarify.
Conclusion
Please write the conclusions more clearly and specifically according to the study findings.
Kind regards
Author Response
The authors would like to express their gratitude to the reviewer for their valuable comments. The authors have addressed and incorporated the reviewer's recommendations and comments into the revised manuscript. The authors' responses have been indicated with "AU." The revised manuscript has been updated to reflect the new suggested changes, which have been highlighted in red font. We hope that the revised manuscript meets the expectations of reviewers and the Journal’s editor.
Dear Authors,
Thank you for the opportunity to review this paper. The study deeply analyses the relationship between polyphenols intake during pregnancy and infant anthropometric measurement. The paper is well written and the statistical methods are robust.
I congratulate the authors for this complex study, including the validation of the FFQ.
AU: We want to thank the reviewer for their comments and kind words regarding our manuscript.
Please find some comments below:
Line 94 The women completed the two FFQs in the first and third trimesters). It is unclear whether the two FFQs are different? Why?
AU: We want to thank the reviewer for their comments and suggestions. In the revised version of the manuscript, line 94 has been modified to provide a more accurate expression. The same FFQ was used in two periods to examine the differences in intake between the first and third trimesters.
Line 137 The foods are classified in groups> cereals and carbohydrates. It is an inappropriate group because cereals are foods and carbohydrates are macronutrients. Please replace somehow.
AU: We would like to express our gratitude to the reviewer for offering constructive feedback. In the updated version of the manuscript, the terms "cereals and carbohydrates" have been replaced with "cereals and derived products".
Line 152-155 It is unclear. The gestational age and anthropometric measurement for newborns…. Please clarify.
AU: As requested, this section of the manuscript has been revised to improve clarity regarding the wording employed.
Conclusion
Please write the conclusions more clearly and specifically according to the study findings.
AU: In accordance with the reviewer's recommendation, the conclusions section has been re-written to enhance clarity in this new version of the manuscript.
Kind regards
AU: The authors would like to express their gratitude to the reviewer for their invaluable feedback and hope that this revised version of the manuscript will be considered for publication.
Reviewer 2 Report
Comments and Suggestions for Authors
The article “Evaluation of polyphenol intake in pregnant women from Southeastern Spain and the effect on anthropometric measures at birth and gestational age” is well written. However I have some hesitations regarding methods of analysis. Below I present some my remarks and suggestions that can improve the manuscript:
1. The Authors wrote that „Flavonoids intake in the 3rd trimester (which represent more than 30% of the total intake) showed a significant direct association with head circumference at birth.” However Figure 2 contradicts such conclusion. We can see (Figure 2) that association of Flavonoids with all anthropometric measures at birth is completely non-significant.
2. Taking into account that reproducibility and validation analysis showed low correlation coefficients for polyphenol intakes as continuous variables, but much higher for categorical (which is typical, of course) why did not you consider to perform the main analysis taking into account quantitative features instead of qualitative?
3. In the discussion the Authors wrote that “Overall, the results showed acceptable validity and high reproducibility across all food groups and polyphenol intake during pregnancy”. Please define in methodology “acceptable validity” and “high reproducibility” – please give the exact cutoff points (tresholds) used for definition with references to the literature. The correlation coefficient were often low, and even negative.
4. Why in Figure 3 only some sub-families of polyphenols were presented? Did you choose only those that were statistically significant with at least one anthropometric measurements?
5. Results for 1st trimester and 3rd trimester are completely different, please explain this and address this problem in the discussion.
6. Please unify in Table 1 format for n (%) and mean +/-SD – (please see variables Maternal age, Parity And Energy Intake (Kcal).
Author Response
The authors would like to express their gratitude to the reviewer for their valuable comments. The authors have addressed and incorporated the reviewer's recommendations and comments into the revised manuscript. The authors' responses have been indicated with "AU." The revised manuscript has been updated to reflect the new suggested changes, which have been highlighted in red font. We hope that the revised manuscript meets the expectations of reviewers and the Journal’s editor.
The article “Evaluation of polyphenol intake in pregnant women from Southeastern Spain and the effect on anthropometric measures at birth and gestational age” is well written. However I have some hesitations regarding methods of analysis. Below I present some my remarks and suggestions that can improve the manuscript:
AU: The authors would like to express their gratitude to the reviewer for their constructive feedback and positive remarks regarding the manuscript. The reviewer's suggestions and comments have proved invaluable in enhancing the revised version of the manuscript, which we hope will meet the reviewer's expectations and represent a more suitable rendering of the original work.
- The Authors wrote that „Flavonoids intake in the 3rd trimester (which represent more than 30% of the total intake) showed a significant direct association with head circumference at birth.” However Figure 2 contradicts such conclusion. We can see (Figure 2) that association of Flavonoids with all anthropometric measures at birth is completely non-significant.
AU: We agree with reviewer 2 that an error was made in the classification of the polyphenol family. As the reviewer correctly noted, the original text mistakenly referred to the family of flavonoids instead of phenolic acids. The description in lines 281-286, which present the subfamily percentages, and the supplementary material, which highlights the significant association, are accurate. This error has been corrected in the revised manuscript, and the manuscript has undergone a comprehensive review to identify any additional errors. Thank you for your careful revision.
- Taking into account that reproducibility and validation analysis showed low correlation coefficients for polyphenol intakes as continuous variables, but much higher for categorical (which is typical, of course) why did not you consider to perform the main analysis taking into account quantitative features instead of qualitative?
AU: We appreciate the reviewer for pointing out this observation. We acknowledge that the reproducibility and validation analysis showed low correlation coefficients, which is not uncommon in nutritional studies. For the shake of transparency and better understanding of the validation procedure, we have expanded Section 2.5 “Data analysis” by including the three methods commonly used in epidemiological studies to demonstrate the robustness of the validation and reproducibility process: a) correlation coefficient (though this is not a measure of agreement but instead a measure of association that can be partly influenced by the size of the sample); b) quintile ranking, in which polyphenol intake was divided into quintiles, and the percentage of data correctly classified in the same or adjacent quintiles was calculated; c) the Bland-Altman method to assess the level of agreement (LOA) between different methods of measurement.
- In the discussion the Authors wrote that “Overall, the results showed acceptable validity and high reproducibility across all food groups and polyphenol intake during pregnancy”. Please define in methodology “acceptable validity” and “high reproducibility” – please give the exact cutoff points (tresholds) used for definition with references to the literature. The correlation coefficient were often low, and even negative.
AU: In the revised manuscript, lines 328-331, we have outlined the threshold values to indicate that both the validation and reproducibility meet the requisite standards, supported by 2 references. Furthermore, section 4.1 outlines additional evaluation and reproducibility studies for both nutrients and polyphenols. A more detailed explanation of the methodology used for the validation and reproducibility has been included in section 2.5 “Data Analysis (See previous comment).
- Why in Figure 3 only some sub-families of polyphenols were presented? Did you choose only those that were statistically significant with at least one anthropometric measurements?
AU: Reviewer 2 is right, we included only the results that were statistically significant. The complete set of results is available in the supplementary material. However, following the reviewer's suggestion, the caption of Figure 3 has been modified to highlight that only statistically significant results are shown.
- Results for 1sttrimester and 3rd trimester are completely different, please explain this and address this problem in the discussion.
AU: As reviewer 2 indicates, the impact of polyphenols varies according to the specific subfamily of polyphenol and the trimester of pregnancy. Accordingly, a clarification has been incorporated into the revised manuscript (lines 439-443).
- Please unify in Table 1 format for n (%) and mean +/-SD – (please see variables Maternal age, Parity And Energy Intake (Kcal).
AU: We would like to thank the reviewer for pointing out this error in the format of Table 1, which has been fixed in the revised version of the manuscript.
Round 2
Reviewer 2 Report
Comments and Suggestions for Authors
The Authors addressed all of my questions, I accept paper in a current form